# Suitability of the VOF Approach to Model an Electrogenerated Bubble with Marangoni Micro-Convection Flow

**Florent Struyven** [1,2,*], **Zhenyi Guo** [3,4], **David F. Fletcher** [3], **Myeongsub (Mike) Kim** [5], **Rosalinda Inguanta** [6], **Mathieu Sellier** [2,*] **and Philippe Mandin** [1,*]

[1] Institut de Recherche Dupuy de Lôme, UMR CNRS 6027, 56100 Lorient, France
[2] Department of Mechanical Engineering, University of Canterbury, Christchurch 8041, New Zealand
[3] School of Chemical and Biomolecular Engineering, The University of Sydney, Sydney, NSW 2006, Australia; zhenyi.guo@outlook.com (Z.G.); david.fletcher@sydney.edu.au (D.F.F.)
[4] Amazon Lab 126, Shenzhen 518000, China
[5] Department of Ocean and Mechanical Engineering, Florida Atlantic University, Boca Raton, FL 33431, USA; kimm@fau.edu
[6] Dipartimento di Ingegneria, Laboratorio di Chimica Fisica Applicata, Viale delle Scienze ed. 6, 90128 Palermo, Italy; rosalinda.inguanta@unipa.it
[*] Correspondence: florent.struyven@univ-ubs.fr (F.S.); mathieu.sellier@canterbury.ac.nz (M.S.); philippe.mandin@univ-ubs.fr (P.M.)

**Abstract:** When a hydrogen or oxygen bubble is created on the surface of an electrode, a micro-convective vortex flow due to the Marangoni effect is generated at the bottom of the bubble in contact with the electrode. In order to study such a phenomenon numerically, it is necessary to be able to simulate the surface tension variations along with a liquid-gas interface, to integrate the mass transfer across the interface from the dissolved species present in the electrolyte to the gas phase, and to take into account the moving contact line. Eulerian methods seem to have the potential to solve this modeling. However, the use of the continuous surface force (CSF) model in the volume of fluid (VOF) framework is known to introduce non-physical velocities, called spurious currents. This paper presents an alternative model based on the height function (HF) approach. The use of this method limits spurious currents and makes the VOF methodology suitable for studying Marangoni currents along with the interface of an electrogenerated bubble.

**Keywords:** electrogenerated bubble; spurious currents; VOF; height function; CFD; Marangoni convection

## 1. Introduction

Bubble management is an important issue during the water electrolysis process for hydrogen or oxygen production. The evolution of bubbles on the electrode surface has multiple effects on the electrochemical process. The bubbles attached to the electrode area raise the overpotential by insulating parts of the electrode surface and the current into the remaining area [1,2], they modify the overpotential concentration by affecting the level of gas supersaturation on the electrode. In addition, attached-bubbles are only produced at active sites on the electrode surface where the gas molecules are supersaturated. Bubbles attached to the surface of the electrodes disturb current distribution and isolate active sites from reaction ions during nucleation and growth preventing other bubbles from being produced. For an effective electrolysis process, the gas released must be promptly removed from the active sites in order to provide more space for the gas release reaction. Consequently, the fast elimination of these bubbles from the electrode is crucial to increase process efficiency and allow the process to operate at higher currents density, which results in higher production rates [3,4]. It is therefore advisable to speed up the detachment of the bubbles from an electrolytic system in order to improve its efficiency.

Usually, a balance of forces acting on the bubble is used to determine the diameter that the bubble will have at the time of detachment and the residence time of the bubble on

the electrode surface [5]. Depending on the experimental conditions, stagnant electrolyte or electrolyte with a flow, vertical or horizontal electrodes, micro or macro electrode, the intensity and direction of the forces involved in the balance may change. The balance of forces generally includes the surface tension force that holds the bubble on the electrode, the buoyancy force, the inertia force, and the contact pressure force that pulls the bubble away from the surface, and the hydrodynamic forces that act in both directions depending on the experimental conditions. As shown in Figure 1, the balance of forces acting on a gas bubble attached to a horizontal electrode include $F_b$ the buoyancy force, $F_i$ the inertial force, $F_d$ the drag force, $F_p$ the contact pressure force, $F_S$ the surface tension force, $F_M$ the Marangoni force, $n$ the normal vector to the electrode. $R_{bubble}$, $R_{contact}$ and $\theta$ the contact angle between the interface and the solid surface are quantities that are involved in the estimation of these forces. The temperature T and concentration c of dissolved species are higher near the bottom of the bubble. The surface tension $\gamma$ varies with the temperature and concentration gradient. This variation is the cause of the Marangoni force.

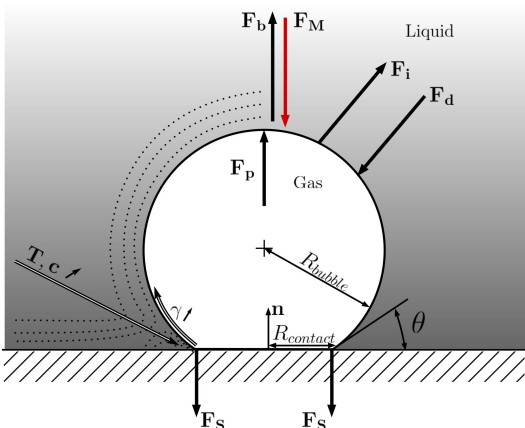

**Figure 1.** Balance of forces acting on a gas bubble attached to a horizontal electrode.

Assuming that the bubble is a sphere, the first approximation of these forces can be expressed as a function of the bubble diameter. In the case of a horizontal electrode, when the projection of this force balance along the normal to the electrode is zero, the bubble detaches from the electrode, which allows us to determine a theoretical diameter of the detachment of the bubble. The bubble detachment diameter is therefore an output of the theoretical model. By comparing it with the experimentally measured one, the accuracy of the theoretical model can be evaluated.

However, if only the previously mentioned forces are considered, the balance of forces cannot predict the bubble detachment diameter. Lubetkin [6] reports different bubble phenomena observed during electrolysis that cannot be explained by the usual force balance. From the study of these phenomena, the author deduced the presence of a force created by the Marangoni effect.

In their 2018 paper, Yang et al. [7] were able to observe a vortex flow at the base of the bubble created by a variation in surface tension along the liquid/gas interface using a particle tracking velocimetry technique (PTV). Yang et al. [7] were able to characterize that this Marangoni effect was created either by a variation of the temperature or by a variation of the concentration of dissolved gas species, acting as a surfactant and reducing the surface tension [8]. Yang et al. [7] also hypothesize that the electrocapillary effect might have an influence on the formation of the vortex. If it is possible to observe these micro-convection phenomena at the scale of a bubble, the clarification of the mechanisms at play remains a scientific challenge that an isolated experimental approach can hardly grasp. Techniques based on high-frequency capture cameras and image processing (PIV, PTV) make it possible to observe the current lines of a liquid at the microfluidic scale, i.e., at the bubble scale. However, they do not allow us to determine the origin of these streamlines. By a single experimental approach, it is not possible to link the intensity or the evolution

of microconvective currents to the variation of the surface tension. Yang et al. [7] could only calculate an approximation of the intensity of these currents. Their local variation as a function of the temperature gradient, chemical species concentration, or voltage cannot be estimated. Moreover several questions need to be addressed regarding the impact of the Marangoni effect on electrogenerated bubbles. How does it affect mass transfer? What is the split of the thermal and solutal Marangoni effect? What about the electrocapillary effect? From this point of view, the contribution of a numerical model could allow validation of hypotheses, or to identify unknown parameters by using inverse techniques.

Among the simulations performed on the subject, Liu et al. were able to simulate the growth of a single hydrogen bubble on the surface of a microelectrode using a VOF method in a two dimensional axisymmetric domain [9]. In their study, the bubble growth rate did not match the experimental results. This can be explained by the fact that the Marangoni effect was not included. In order to compare their experimental observations with a numerical model Massing et al. were able to simulate the Marangoni effect along a with fixed interface [10].

Liu et al. [9] use a multiphase model for which the surface tension force is calculated from the CSF model. In the simulations of Massing et al. [10], the surface tension force is represented by a boundary condition on a fixed interface, i.e., assumed to be solid. Consequently, the answers provided by these simulations remain limited and do not allow the complete phenomenon to be understood. Simulating the growth of a bubble without the Marangoni effect is erroneous, and studying the Marangoni effect over a limited time without including the deformation of the bubble does not allow understanding of the evolution of the concentration profile of the dissolved species taking place on the periphery of the bubble. The coupling of mass transfer on a free interface on which the Marangoni effect is simulated is necessary.

According to recent observations by Yang et al. [7] on a microelectrode and Lubetkin's assumptions [6] a Marangoni current can develop along the interface of the bubble during its growth. However, there is no consensus on the origin of this Marangoni effect and its intensity. The Marangoni effect could be solutal or thermal in origin. To be able to perform a direct simulation of the phenomenon it is necessary to be able to couple the interfacial mass transfer and the surface tension variations along the interface to simulate the Marangoni effect. The Marangoni effect can potentially influence bubble growth by intensifying the mass transfer of dissolved gases from the electrode surface to the bubble interface. Another assumption to consider is that mass transfer is greatest at the contact line between the bubble surface and the electrode surface, i.e., where the concentration of dissolved gases is greatest. To be able to simulate what is described above throughout the growth of the bubble, it is necessary to be able to include simultaneously in a numerical model the interfacial mass transfer, the variations of surface tension, and to follow the displacement of the contact line.

In order to accommodate the needs of such a simulation a holistic model is required. The model used must be multiphase, multiphysics to take into account the transport of dissolved species, must include an efficient surface tension model able to take into account the effects of a variable surface tension, must simulate the mass transfer across the interface, must be able to simulate the moving contact lines. Moreover, the chosen model must be able to avoid the main errors related to multiphase and microfluidic models, such as spurious currents and volume conservation problems [11].

In his review Wörner [11] inspects the methodologies for modeling multiphase fluids at the microfluidic scale. He presents studies based on Eulerian and Lagrangian representations. The choice of an efficient and robust method to take into account the interface depends on the physical problem to be studied, as each method has its strengths and weaknesses. If solutions are presented to model the mass transfer, the Marangoni effect or to reduce the spurious currents, to the best of the author's knowledge, no current solution allows the coupling of these two features, i.e., modeling the Marangoni effect on a free interface including mass transfer. The choice of different methods to perform these functions

individually is based less on their individual performance but more on their compatibility. In this spirit, a method based on VOF and height functions seems a good compromise.

VOF methods naturally ensure the conservation of volume and mass in incompressible flows and, with some improvements, in compressible flows. However, the description of the interface is diffuse, which makes it difficult to evaluate the curvature of the interface and impose boundary conditions. The level-set method, like the VOF methods, automatically takes into account topological changes. It describes the interface implicitly using a signed distance function which gives a more precise definition of the interface than in standard VOF methods. But the signed distance function must be reset frequently violating mass conservation. Finally, the Lagrangian methods are very precise with a thickness free interface and boundary conditions that are imposed exactly on the interface. However, changes in topology and highly deformed interfaces are not easily implemented in these methods because the remeshing procedure which maintains an adequate mesh size can become very complicated in this case.

Spurious currents are artificially created by the numerical model used to calculate the pressure jump between each phase resulting from the surface tension force. At the microfluidic scale, these artificial currents cannot be neglected because their intensity competes with that of the real currents that are the focus of the numerical simulation, i.e., in our case, the Marangoni currents. By its nature, the VOF method has good volume conservation properties [11]. In other words, after the advection step performed by the solver, the sum of the volumes of each phase present in each cell of the mesh is correctly preserved, i.e., no volume of a phase is artificially created or removed by numerical errors. Various surface tension models have been adapted to reduce spurious currents, including methods based on height functions [12–14]. The height function method allows simulation of the effects of Marangoni of solutal or thermal origin [15], and can also be adapted to solve contact line problems [16–18]. The VOF method is also well-adapted to simulate mass transfer physics for micro-scale simulations [19].

Taqiedin et al. [5] explored different methodologies for simulating electrogenerated bubbles. Depending on the problem and the topology of the interface, a large density ratio between phases can lead to significant performance degradation. Popinet [12] and Abadie et al. [20] studied density ratios close to 1000, but the case of a hydrogen bubble in an electrolyte is an extreme case with a density ratio close to more than 10,000 and therefore should be tested. Moreover, since spurious currents are inversely proportional to the capillary number $Ca \approx 4 \cdot 10^{-4}$ (with $Ca = \mu V_{char} \gamma^{-1}$), i.e., depending on the fluid velocity, it is important to perform specific tests for the system under study.

In this perspective, the novelty of this study is to integrate the height functions methodology with the methodologies used by Seric et al. [15] and Soh et al. [21] and to test it in the specific case of electrogenerated bubbles, i.e., with low capillary number and high density ratio. A new methodology to calculate the interface density in each cell of the mesh has been developed accordingly. The goal is to evaluate the error that the spurious currents will generate on the final calculation of the Marangoni currents according to the methodology described in the following article. For comparison, these results will be compared with the CSF model. The two test cases used allow characterization of the spurious currents and to simulate as close as possible the conditions of the system in which an electrogenerated bubble on a microelectrode can be observed.

## 2. Mathematical Model

In a monofluid system, the conservation principle applied to mass and momentum is correct, and the localization principle allow us to establish the Navier-Stokes equations. These same principles applied to two-fluid systems allows us to find the Navier-Stokes equations but with the addition of a jump relation to describe the exchange of mass and momentum at the interface between the two fluids. A diagram of a small volume consisting of a gas and liquid phase is shown in Figure 2.

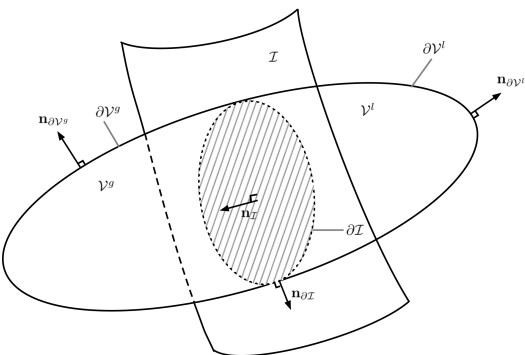

**Figure 2.** Diagram of a fluid particle. The gaseous $\mathcal{V}^g$ and liquid $\mathcal{V}^l$ volumes are separated by an interface $\mathcal{I}$.

Before the conservation principle can be applied to the two-fluid system, a jump operator must first be defined. This operator describes the passage of a quantity $Q$ through an interface between two distinct areas. Assuming that $Q$ has a limit on each side of the interface $\mathcal{I}$, we define for any point $\mathbf{x}$ belonging to $\mathcal{I}$, the jump relation for $Q$ by:

$$[\![Q]\!] = \lim_{h \to 0^+} \left[ Q^l(\mathbf{x} + h\, \mathbf{n}_{\mathcal{I}}) - Q^g(\mathbf{x} + h\, \mathbf{n}_{\mathcal{I}}) \right] \tag{1}$$

where $h$ is a scalar, $\mathbf{n}_{\mathcal{I}}$ is the unit vector normal to the interface and the superscripts $l$ and $g$ represent the liquid and gas phases, respectively.

In order to take into account the Marangoni effect it is necessary to define the surface gradient operator $\nabla_s Q$. It is defined as the projection of the gradient $\nabla Q$ onto the surface:

$$\nabla_s Q = \nabla Q - \mathbf{n}_{\mathcal{I}}(\mathbf{n}_{\mathcal{I}} \cdot \nabla Q) \tag{2}$$

In each phase the principle of conservation of mass and momentum applies:

$$\nabla \cdot \mathbf{v} = 0 \tag{3}$$

$$\frac{\partial \rho \mathbf{v}}{\partial t} + \nabla \cdot (\rho\, \mathbf{v} \otimes \mathbf{v}) = \nabla \cdot (\mathbb{T}) + \rho \mathbf{f_v} \tag{4}$$

the stress tensor can be decomposed into a pressure component and a viscous stress tensor:

$$\mathbb{T} = -p\mathbb{I} + 2\mu\mathbb{D} \tag{5}$$

where $\mu$ is the dynamic viscosity, $\mathbb{D} = \left(\nabla\mathbf{v} + (\nabla\mathbf{v})^T\right)/2$, and $\mathbf{f_v}$ are the volumetric forces.

At the interface $\mathcal{I}(t)$ for mass conservation the jump relation is:

$$[\![\rho\,(\mathbf{v} - \mathbf{v}_{\mathcal{I}})]\!] \cdot \mathbf{n}_{\mathcal{I}} = 0 \tag{6}$$

where $\mathbf{v}_{\mathcal{I}}$ is the velocity of the interface. This last relation reflects the equality of the mass flows on each side of the interface. By this means an input parameter of the model is introduced, the mass flow rate of mass transfer at the interface $\dot{m}$ in $[\mathrm{kg} \cdot \mathrm{m}^{-2} \cdot \mathrm{s}^{-1}]$.

$$\dot{m} = \rho^g\,(\mathbf{v}^g - \mathbf{v}_{\mathcal{I}}) \cdot \mathbf{n}_{\mathcal{I}} = \rho^l\,(\mathbf{v}^l - \mathbf{v}_{\mathcal{I}}) \cdot \mathbf{n}_{\mathcal{I}} \tag{7}$$

At the interface for momentum conservation:

$$[\![\rho\mathbf{v} \otimes (\mathbf{v} - \mathbf{v}_{\mathcal{I}})]\!] \cdot \mathbf{n}_{\mathcal{I}} - [\![\mathbb{T}]\!] \cdot \mathbf{n}_{\mathcal{I}} - \gamma\,\kappa\mathbf{n}_{\mathcal{I}} - \nabla_s \gamma = 0 \tag{8}$$

Applying mass conservation at the interface $[\![\rho\,(\mathbf{v} - \mathbf{v}_\mathcal{I})]\!] \cdot \mathbf{n}_\mathcal{I} = 0$ and so $[\![\rho\mathbf{v} \otimes (\mathbf{v} - \mathbf{v}_\mathcal{I})]\!] \cdot \mathbf{n}_\mathcal{I}$ can be expressed as $\dot{m}[\![\mathbf{v}]\!]$. At the interface by using the mass flow rate and developing the stress:

$$\dot{m}[\![\mathbf{v}]\!] + [\![p\mathbb{I}]\!] \cdot \mathbf{n}_\mathcal{I} - [\![2\mu\mathbb{D}]\!] \cdot \mathbf{n}_\mathcal{I} = \gamma\kappa\mathbf{n}_\mathcal{I} + \nabla_s\gamma \tag{9}$$

Projections along a normal axis and an axis tangential to the interface results in:

$$\dot{m}[\![\mathbf{v}]\!] \cdot \mathbf{n}_\mathcal{I} + [\![p]\!] - [\![2\mu\mathbb{D} \cdot \mathbf{n}_\mathcal{I}]\!] \cdot \mathbf{n}_\mathcal{I} = \gamma\kappa \tag{10}$$

$$\dot{m}[\![\mathbf{v}]\!] \cdot \mathbf{t}_\mathcal{I} - [\![2\mu\mathbb{D} \cdot \mathbf{n}_\mathcal{I}]\!] \cdot \mathbf{t}_\mathcal{I} = \nabla_s\gamma \cdot \mathbf{t}_\mathcal{I} \tag{11}$$

Equations (10) and (11) help understand how surface tension, the Marangoni effect, and mass transfer influence the flow at the interface.

This article proposes to test the surface tension model on two test cases. The first test case that we consider is the 2-dimensional static bubble, the second consists of a uniform flow field that translates the bubble as proposed by Popinet [12]. By applying the hypotheses related to each of these systems, the Navier-Stokes equations are simplified: Newtonian, considered incompressible, no mass transfer at the interface, no gravity, constant surface tension, the flow is isothermal.

The volumetric forces term cancels out in the momentum equation. In the two test cases, since there is no mass transfer at the interface, the velocity of the two fluids on either side of the interface is equal to that at the interface. For the first test case starting from Equation (10), and assuming $\mathbf{v} = \mathbf{0}$ on either side of the interface, the Young-Laplace equation $-[\![p]\!] = \gamma\kappa$ appears. The two phases are in equilibrium, which theoretically means that the velocity is zero on both sides of the interface. This means that if the fluid is moving near the interface it can only be due to a numerical error.

For the second case the normal projection of the momentum jump relation is reduced to:

$$[\![p]\!] - [\![2\mu\mathbb{D} \cdot \mathbf{n}_\mathcal{I}]\!] \cdot \mathbf{n}_\mathcal{I} = \gamma\kappa \tag{12}$$

The first and second terms in the preceding equation represent the pressure jump and stress tensor at the interface, respectively, which are in equilibrium with the third term due to the surface tension effect.

## 3. Spurious Currents

Instead of considering two fluids (gas-liquid), the VOF method assumes a single fluid, and solves a single conservation equation for mass and momentum. A continuous indicator function $\mathbb{1}$ is used to distinguish the liquid phase from the gas phase. It takes the value 1 when the liquid phase is present at a given point of the system and 0 otherwise. In the VOF methodology, the integral over the volume of the mesh cell of the indicator function defines the volume fraction $\alpha$:

$$\alpha = \frac{1}{V_{cell}} \int_{V_{cell}} \mathbb{1}\,dV \tag{13}$$

One of the most common surface tension models used for VOF is the continuum surface force model CSF. By using a Dirac delta function, the surface tension expressed through the jump relation is transformed into a volumetric force. This volumetric formulation makes it possible to integrate surface tension as a source term in the momentum equation. Locally in a small volume, we get:

$$\mathbf{f_{fl}} = \gamma\kappa\mathbf{n}_\mathcal{I}\delta_\mathcal{I} \tag{14}$$

where $\delta_\mathcal{I}$ is the dirac function which is non-zero only on the interface. The geometrical properties of the interface, the normal vector and the curvature can be calculated from the gradient of $\alpha$.

$$\mathbf{n}_\mathcal{I} = \frac{\nabla\alpha}{|\nabla\alpha|} \tag{15}$$

$$\kappa = \nabla \cdot \mathbf{n}_{\mathcal{I}} = \nabla \cdot \left( \frac{\nabla \alpha}{|\nabla \alpha|} \right) \tag{16}$$

when discretizing and choosing the surface tension model, one must ensure:

- the existence of mechanical equilibrium when the fluids are at rest;
- the normal unit vector and curvature are estimated accurately;
- the coherence of the discretization operators.

It is assumed that there is a discretized interface geometry such that a zero velocity field is the solution to the Navier-Stokes equations. The mechanical equilibrium of the discrete system at zero velocity is characterized by the momentum equation, where all velocity-dependent terms are removed. This leaves a balance between the discrete pressure gradient, and the surface tension term. Popinet [22] specifies that for the equilibrium condition to be met, the pressure gradient should be estimated using the same discrete operator as that used to estimate the gradient of the indicator function used in the volumetric surface tension force calculation. He points out that Brackbill et al. [23] in the original CSF article uses two different operators to calculate the pressure gradient whose values are taken from the centre of the faces and the gradient of the volume fraction whose values are taken from the centre of the cells. They calculate the surface tension force on the centre of the faces by averaging the values taken at the centre of the cell to perform the force balance. The values used to calculate the discretized gradient of pressure and volume fraction must be taken at the same location in the mesh in order to get a well-balanced relation. If this is not the case, the discrete operators used to calculate the pressure gradient and volume fraction are not the same and an imbalance is created at the time of the equilibrium. However, the use of this discrete operator does not give a sufficient approximation of the gradient of the volume fraction to be able to estimate the interface normal and the curvature accurately. Popinet [22] suggests calculating the curvature using another method than that used in the CSF method, i.e., without using the gradient of the volume fraction.

As shown in Figure 3 the spurious currents are not negligible in the case of the CSF method. For the height function method they can be reduced by refining the mesh. The normal component of the surface tension was calculated with the CSF methodology for the simulation of the right image, and the height function and polynomial fit methodology for the simulation of the left image. For both simulations the tangential component of the surface tension was calculated from Equation (28). In the case of the CSF methodology the spurious currents distort the calculation of the Marangoni currents, which is not the case of the methodologies used in the simulation of the left image. In order to obtain an accurate calculation of the Marangoni currents it is important to use a methodology that limits spurious currents.

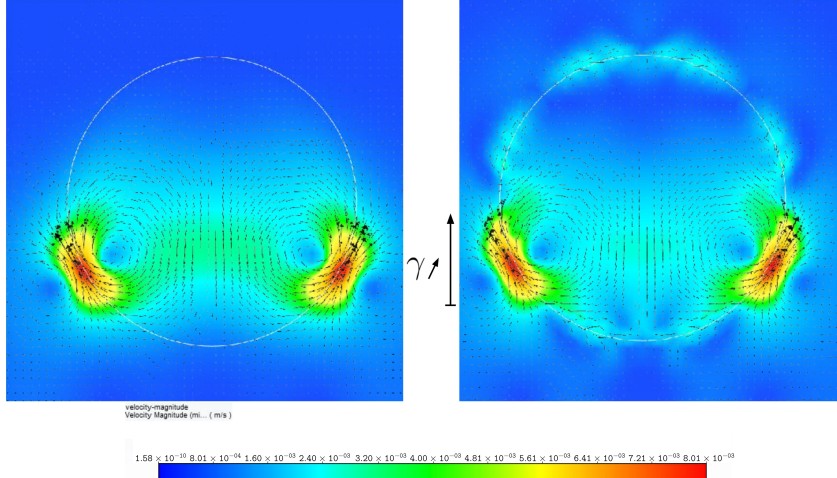

**Figure 3.** Example of a simulation, with a Marangoni current generated along with the interface of a bubble from a gradient of surface tension.

## 4. Surface Tension Model

The height function method makes the calculation of normal and curvature more accurate, thus reducing parasitic currents [12,13,24–26]. It can be integrated into surface tension models, such as the Brackbill's model. An interface can always be described locally as a graph of a function. The principle of the height function method is to use the local coordinate system to be able to find the curvature of the interface. As illustrated in Figure 4, by calculating the integral of this function and dividing it by the interval over which it has been integrated, we obtain the mean value or height $H$ of this function over the interval, [13]. The volume fraction gives the part of the cell area occupied by a phase. A stencil of several cells around the cell through which the interface passes and for which the curvature is to be calculated is used as the basis for a coordinate system. In each cell of width $\Delta x$ and height $\Delta y$ we have the value of the volume fraction occupied by one of the phases. Summing the volume fractions of a column of the stencil and multiplying it by the width of the cells makes it possible to carry out a calculation equivalent to the calculation of an integral of a function whose graph is represented by the interface in a local coordinate system represented by the stencil:

$$H(x_0) = \int_{x_0 - \frac{\Delta x}{2}}^{x_0 + \frac{\Delta x}{2}} f(x) / \Delta x \, dx \tag{17}$$

$$H(i) = \sum_{j=-\infty}^{j=+\infty} \alpha_{ij} \, \Delta y \tag{18}$$

$$H(x_0) = H(i) \tag{19}$$

where $i$ and $j$ are respectively the horizontal and vertical indexes identifying the cell in the mesh, and $x_0$ is the abscissa corresponding to index $i$ in the locally defined coordinate system. From this quantity we can obtain an approximation for the first and second derivatives of the function in the $i$th column by using the value of the height in left $(i - 1)$ and the right $(i + 1)$ column of the stencil.

$$H'(i) = \frac{H(i+1) - H(i-1)}{2\Delta x} \tag{20}$$

$$H''(i) = \frac{H(i+1) - 2H(i) + H(i-1)}{\Delta x^2} \tag{21}$$

an estimation of the tangential vector, the normal vector, and the curvature can be obtained from there:

$$\mathbf{t}_\mathcal{I} = \frac{1}{\sqrt{1 + H'(i)^2}} \begin{pmatrix} 1 \\ H'(i) \end{pmatrix} \tag{22}$$

$$\mathbf{n}_\mathcal{I} = \frac{1}{\sqrt{1 + H'(i)^2}} \begin{pmatrix} H'(i) \\ 1 \end{pmatrix} \tag{23}$$

the curvature is calculated as the negative of the divergence of the normal vector:

$$\kappa = -\nabla \mathbf{n}_\mathcal{I} = -\frac{H''(i)}{\left(1 + H'(i)^2\right)^{3/2}} \tag{24}$$

the key to the success of this method is having access to sufficiently accurate discrete values for the height function [27]. So, the smaller the cell width, the better the approximation. An issue arises when the slope of the interface tends to infinity. In mathematical terms, the function is no longer defined. The more the interface is tilted with respect to the mesh axes, the less accurate the curvature computation with the height functions will be, as already noted by Popinet [12]. When the interface is diagonal to the axes of the mesh, another method must be used.

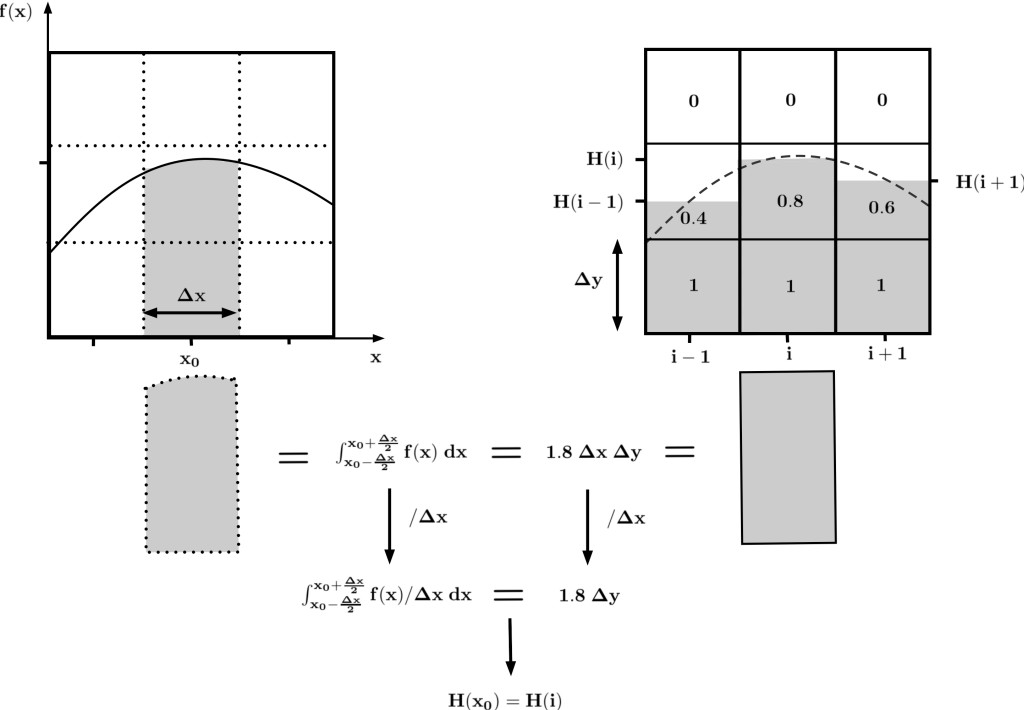

**Figure 4.** Methodology of height functions. The sum of the volume fractions present in a column is equal to the average value of a function $f$ on an interval $\Delta x$ of which the curve represents the interface.

The interface normal approximated with the height function method is used as a starting point to construct a new cell-specific reference frame for the cell for which we want to calculate the curvature. A first calculation is performed to determine the coordinates of the barycentre of the reconstructed interface fragment contained in the cell. This barycentre and the unit vector normal to the interface allow us to define an orthonormal coordinate system. The coordinates $x_{\mathcal{I}}$ of the $n$ positions of the interface estimated previously with the calculation of the heights are calculated in the new reference frame. By minimizing the objective function $f_{obj}$, the parameters $p_i$ of the parabola that best approximates the interface are estimated.

$$f_{obj}(p_i) = \sum_{i=1}^{3} |x_{\mathcal{I}} - f_{par}(p_i, x_{\mathcal{I}})|^2 \tag{25}$$

with

$$f_{par}(p_i, x_{\mathcal{I}}) = p_0 x_{\mathcal{I}}^2 + p_1 x_{\mathcal{I}} + p_2 \tag{26}$$

the mean curvature at the origin $o$ is then calculated:

$$\kappa = \frac{2p_0}{(1 + p_1^2)^{3/2}} \tag{27}$$

It is emphasized by Seric et al. [15] that using the surface gradient operator as it is defined in the continuum surface force model can result in inaccuracies when implemented in the VOF method. One of the disadvantages of the VOF method is that the described interface is diffuse. The Marangoni effect is caused by tangential stress located on the interface due to a variation in surface tension. Surface tension is a concept that only makes physical sense at the interface and its variation only makes sense along that interface. An estimation of the surface tension gradient is necessary to be able to calculate the surface gradient operator. The surface tension is a function of, among other things, the temperature and the concentration of chemical species. However, these quantities generally vary abruptly when they are considered on either side of the interface.

Therefore, in VOF and similar methods, the gradient of surface tension is a vector whose direction is close to the normal of the interface, which makes no physical sense. The tangential component is obtained by subtracting the normal component from the surface gradient operator. This approximation constructed from data from a few cells (depending on how the gradient operator is calculated) is very sensitive to errors.

Seric et al. [15] propose to implement the variation of surface tension using a method inspired by height functions. In this method, the derivative along the interface of the surface tension is calculated directly:

$$\mathbf{f_M} = \frac{\partial \tilde{\gamma}}{\partial s} \, \delta_s \mathbf{t}_\mathcal{I} \tag{28}$$

where $s$ is the arc length. The derivative is computed as:

$$\left( \frac{\partial \gamma}{\partial s} \right)_{i,j} = \frac{\tilde{\gamma}(i+1) - \tilde{\gamma}(i-1)}{ds} \tag{29}$$

$\tilde{\gamma}(i)$ is the weighted average of the values taken by the surface tension for the cells of column $i$. Therefore the same value of surface tension will be used in the calculation of the derivative for the cells of the same column of the stencil.

$$\tilde{\gamma}(i) = \frac{\sum\limits_{j=-\infty}^{j=+\infty} \alpha_{ij} \, \gamma_{ij}}{\sum\limits_{j=-\infty}^{j=+\infty} \alpha_{ij}} \tag{30}$$

then the derivative of the arc length is calculated from the derivative of the height function:

$$ds = 2\Delta_x \sqrt{1 + H'(i)} \tag{31}$$

In this method the tangential component at the interface of the surface tension gradient is directly obtained which avoids the projection along the tangent to the interface thus avoiding the calculation errors generated by the surface gradient operator. Its other advantage is that the diffuse data due to the VOF method are averaged in a direction close to the normal to the interface, the columns of the stencil being oriented in the direction almost perpendicular to the interface.

In the literature, a common method for determining $A_\mathcal{I}$ is to calculate $|\nabla \alpha|$.

$$A_\mathcal{I} = \int_{\mathcal{I}_{cell}} dS = |\nabla \alpha| V_{cell} \tag{32}$$

In three spatial dimensions, the volume integral gives the interface area, in two dimensions the corresponding area integral gives the interface length. Note that this function is non-zero only at the interface between the two phases. It correctly represents the interfacial surface between the phases over the whole computational domain. Nevertheless, the adoption of such a function to approximate the interfacial area is not without limitations. It can be shown that the method is only valid at a global level and that the function does not adequately represent the interfacial area locally [19,28]. It can have non-zero values for cells directly adjacent to the interface. Considering $\mathbf{f_{fl}}(i,j)$ the source term assigned to the index cell $(i,j)$, in each cell we can discretize the force that will balance the momentum equation.

$$\mathbf{f_{fl}}(i,j) = \gamma \, \kappa(i,j) \frac{A_\mathcal{I}(i,j)}{V_{cell}} \mathbf{n}_\mathcal{I}(i,j) \tag{33}$$

The use of the volume fraction gradient is a source of error. With the developments made previously on the height functions it is possible to calculate $A_\mathcal{I}$ without using the gradient of the volume fraction. For a 2D simulation, by determining the coordinates of the

points where the interface crosses the axes of the mesh and knowing $f_{par}$ the length of the interface can be determined by using Algorithm 1 allowing estimation of the length of the curve of a function.

---

**Algorithm 1:** Calculation of $A_{\mathcal{I}}$ in 2D

---

**Result:** return $A_{\mathcal{I}}$
$n_{iter} = 0$
$A_{\mathcal{I}} = 0$
$step_x = \frac{|x_2 - x_1|}{n_L}$
$x_a = 0, \ x_b = x_a + step_x$
**while** $n_{iter} < n_L$ **do**
$\quad\mid\quad L = 0$
$\quad\mid\quad y_a = f_{pol}(x_a)$
$\quad\mid\quad y_b = f_{pol}(x_b)$
$\quad\mid\quad L = \sqrt{(x_b - x_a)^2 + (y_b - y_a)^2}$
$\quad\mid\quad A_{\mathcal{I}} = A_{\mathcal{I}} + L$
$\quad\mid\quad x_a = x_a + step_x$
$\quad\mid\quad x_b = x_b + step_x$
$\quad\mid\quad n_{iter} = n_{iter} + 1$
**end**

---

Meier et al. used a similar method [29]. The calculation of $A_{\mathcal{I}}$ can be extended to a 3D simulation [21]. The use of such a computational method is necessary to be able to integrate a mass transfer model across the interface [19].

For the mass transfer rate to be calculated accurately $A_{\mathcal{I}}$ must be evaluated correctly [21,28]. The gradient of $\alpha$ gives a biased representation of the interface. At the local level, $|\nabla\alpha|$ can have non-zero values in cells of the mesh where in the continuous model the interface is not present. These cells are adjacent to the interface cells, $|\nabla\alpha|$ is computed using the $\alpha$ values of the neighboring cells, including the interface cells. Thus $|\nabla\alpha|$ may not be zero for cells where $\alpha = 0$ and $\alpha = 1$. These cells can generate an artificial mass transfer and after the transfer phase the calculation can result in a value of $\alpha$ that is negative or greater than unity.

## 5. Solver Settings

The commercial software Ansys Fluent 2020 R2 was used to perform the numerical simulations. The model was coded and implemented using User-Defined Functions (UDFs).

To ensure that spurious currents do not develop over time for explicit schemes, a stability condition on the time step introduced by Brackbill et al. [23] must be applied:

$$\Delta t < \sqrt{\frac{\rho_{avg}(\Delta x)^3}{2\,\pi\gamma}} \tag{34}$$

where $\Delta x$ is the grid spacing, $\gamma$ the surface tension, $\rho_{avg}$ is the average density of the phases. The physical reason given by Brackbill et al. [23] is that the time step must be small enough to resolve the fastest capillary waves. The value of the time step is limited by the size of the mesh. As shown by equation (34) there is a power law relationship between the time step and the grid spacing, $\Delta t \propto (\Delta x)^{3/2}$. See [22] for a detailed discussion on the subject.

The mesh consists of square and orthonormal cells. Different mesh sizes were used for the simulation: $40 \times 40$, $80 \times 80$, $120 \times 120$, $160 \times 160$. The gradients of scalars are calculated as cell centroid values from the centroid values of faces surrounding the cell. The Green–Gauss node-based method is used for this calculation. The PRESTO scheme is used for pressure interpolation. The QUICK scheme is used for the discretisation of the momentum and the energy equations. The Piecewise-Linear Interface Calculation (PLIC) scheme is used for the discretisation of the volume fraction equation. When simulations are

made using the CSF method, the default node based smoothing of the volume fraction field prior to calculation of the curvature was enabled. No smoothing of the calculated curvatures was performed. A first order implicit scheme is used for the temporal discretisation of the transient terms. Finally, for the pressure–velocity coupling, the SIMPLE algorithm is used. Globally scaled residuals are used and the residual targets for all the equations are set to $1 \times 10^{-6}$. The properties of the electrolyte and the gas considered ($H_2$) are shown in the Table 1.

**Table 1.** Physical parameters used.

| $\rho_l \, [\mathbf{kg \cdot m^{-3}}]$ | $\rho_g \, [\mathbf{kg \cdot m^{-3}}]$ | $\mu_l \, [\mathbf{kg \cdot m^{-1} \cdot s^{-1}}]$ | $\mu_g [\mathbf{kg \cdot m^{-1} \cdot s^{-1}}]$ |
|:---:|:---:|:---:|:---:|
| 1000 | 0.0899 | $1.2 \times 10^{-3}$ | $8.79 \times 10^{-6}$ |

As noticed by Lubetkin the surface tension varies with the concentration of dissolved gases and the temperature [6]. To be consistent with the experiments performed on microelectrodes, the initial surface tension was set at 0.075 $[\mathrm{N \cdot m^{-1}}]$ [9,30]. In our case and according to Vasquez et al. [31] it varies with temperature linearly:

$$\frac{\partial \gamma}{\partial T} = -1.6 \times 10^{-4} [\mathrm{N \cdot m^{-1} \cdot K^{-1}}] \tag{35}$$

The linear temperature variation can be approximated to about ten degrees. The electro-capillary effect is a less well understood phenomenon in the context of electrolysis. The value of $\frac{\partial \gamma}{\partial \Phi}$ is not agreed upon and can vary according to the authors between $2 \cdot 10^{-3}$ and $10^{-7}$ $[\mathrm{C \cdot m^{-2}}]$ [32–34].

Similarly, the work of Massoudi et al. [8] has been able to establish relationships between the variation in partial pressure of gas and surface tension. At low pressures, the concentration of dissolved hydrogen and the partial hydrogen pressure can be related through Henry's law:

$$c_{h_2} = p \, K_H \tag{36}$$

where $K_H$ the constant of proportionality is dependent on the temperature and pressure, but in our case can be estimated as [35,36]:

$$K_H = 7.8 \times 10^{-6} \, [\mathrm{mol \cdot m^{-3} \cdot Pa^{-1}}] \tag{37}$$

As a result, the variation of surface tension as a function of concentration can be obtained:

$$\frac{\partial \gamma}{\partial c_{H_2}} = \frac{1}{K_H} \frac{\partial \gamma}{\partial p} = -3.2 \times 10^{-5} \, [\mathrm{N \cdot m^3 \cdot m^{-1} \cdot mol^{-1}}] \tag{38}$$

By relating this coefficient to the $H_2$ saturation concentration $c_{H_2}^s = 0.75 \times 10^{-3} \, [\mathrm{mol \cdot L^{-1}}]$, it can be seen that the variation of the surface tension considered in the physical problem is negligible compared with the numerical problem. However, these small variations in surface tension are sufficient to generate Marangoni currents at the scale of the bubble [7,10]. They reach velocities of up to $u_M \approx 20 - 30 \, [\mathrm{mm \cdot s^{-1}}]$. These Marangoni currents are the main focus of this study. This order of magnitude obtained experimentally is to be compared with the intensity of the spurious currents.

## 6. Curvature Error Calculation

It has been shown by Cummins et al. [24] that starting from an exact volume fraction value, calculations with the standard method of height functions estimate the curvature asymptotically with second order accuracy. As previously mentioned, the height function method loses its effectiveness when the interface approaches the diagonal of the mesh axes. In order to test the efficiency of the method used, several tests have been performed for different spatial resolutions. The curvature calculation tests were performed on circular interfaces. The evaluated curvature is compared with the exact curvature. To prevent the

curvature calculation from being compromised by errors in the numerical calculation of the volume fraction, the curvatures were evaluated from analytically calculated exact volume fractions, so that only the curvature calculation method is evaluated:

$$\Delta \kappa_{error} = \frac{1}{\kappa_{exact}} |\kappa - \kappa_{exact}| \tag{39}$$

Curvatures are evaluated for interfaces with a tangent inclined at an angle $\theta$ of 0 to 45° with respect to the horizontal axis of the mesh.

The usual height function method as mentioned before presents good results for interfaces whose inclination is close to the axes of the mesh. However, from a certain inclination, the results obtained diverge from the real value of the curvature, as shown in Figure 5. The choice of the alternative method of polynomial fit is to be considered. In view of the results obtained, the transition from one method to the other is applied for an interface tilt around $\theta_{switch} = 22.5°$. The main difficulty of the polynomial fit method is to choose the interface points to use. When $\theta$ is less than $\theta_{switch}$, the height function method is used. The transition to the polynomial fit method is made for $\theta$ greater than $\theta_{switch}$. With regard to the results presented in Figure 6 for resolutions where the radius of the circular interface considered has a length equivalent to 5 cell widths of the mesh, it appears that the use of the two techniques is inconsistent. The polynomial fitting method is not accurate enough. The points approximating the position of the interface are not close enough to be able to correctly estimate the polynomial parameters. On the other hand, when the spatial resolution becomes finer and for $\theta$ greater than $\theta_{switch}$ the fitting method gives better results.

These tests on the curvature allowed us to establish the best value for $\theta_{switch}$. The center of the circular interface was moved to test the robustness of the methodology. In general this test has no influence on the results obtained. However, in cases where the part of the interface present in the cell is too small, the calculation error on the curvature diverges. The choice of the weighted average calculation of the curvatures on the adjacent cells was considered instead. This choice allows much better results to be obtained. The electrogenerated bubbles having almost circular interfaces so the choice of an average seems coherent. Generally speaking, for resolutions for which the interface radius is equivalent to 15 times the width of the mesh, calculation errors of less than 0.3% are obtained. This preliminary test allows us to be confident about the curvature calculation methodology.

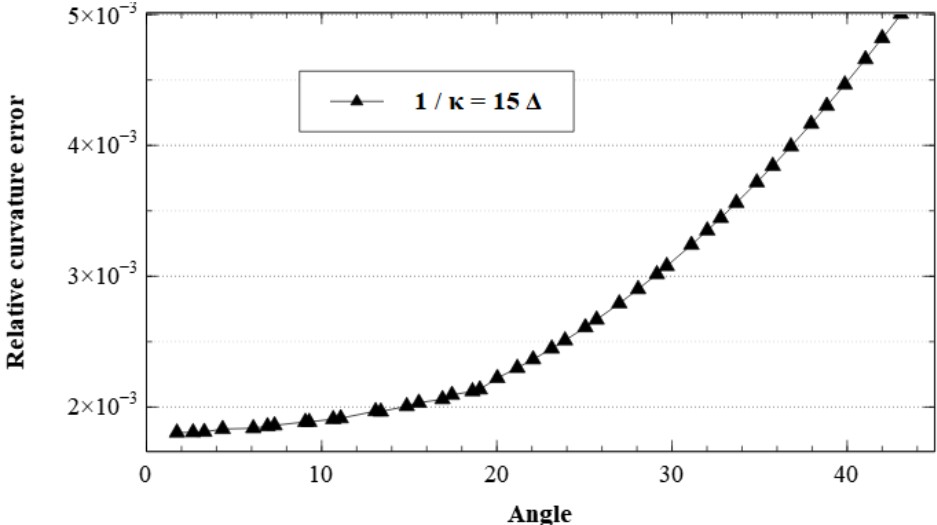

**Figure 5.** Relative curvature errors with the height function methodology along the circular interface as a function of the interface inclination angle with respect to the horizontal.

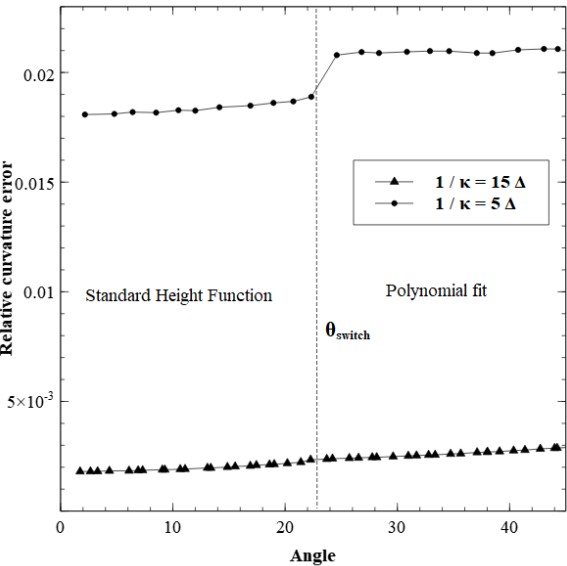

**Figure 6.** Relative curvature errors along the circular interface as a function of the interface inclination angle with respect to the horizontal for two interface resolutions.

## 7. Interfacial Area Error Calculation

In order to evaluate the accuracy of the calculation of the interfacial area of each cell of the mesh, several simulations were conducted. The calculated value is compared with the exact value of the interface $A_{\mathcal{I},exact}$. The test case of a static bubble as described below was used. The interface being circular the exact value of the interfacial area in each cell can be calculated analytically. As the position of the interface in each cell can influence the calculated numerical value, the values of all cells through which the interface passes were averaged, as described by the following equation:

$$E(A_{\mathcal{I}}) = \frac{\sum\limits_{}^{N}|A_{\mathcal{I},exact} - A_{\mathcal{I}}|}{N} \qquad (40)$$

where $N$ is the number of cells for which the calculation was performed. The results are shown in the graph in Figure 7.

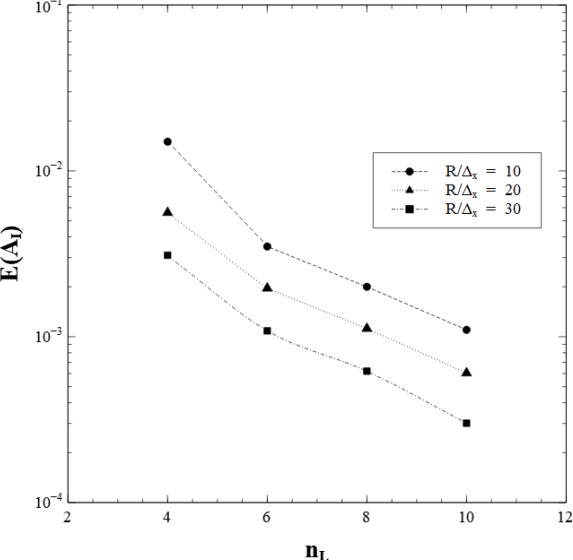

**Figure 7.** Relative interfacial area errors averaged along a circular interface as a function of $n_L$ and the mesh refinement.

The parameter $n_L$ determines the number of segments for which the interface length in each cell of the mesh is cut. The calculations were performed as a function of the parameter $n_L$, which determines the accuracy of Algorithm 1, and as a function of the ratio $R/\Delta_x$, which determines the accuracy of the mesh.

The graph in Figure 8 shows the maximum error found, as described by the following equation:

$$E_{\max}(A_\mathcal{I}) = \max|A_{\mathcal{I},exact} - A_\mathcal{I}| \tag{41}$$

the error decreases with a finer mesh, and by increasing the value of the parameter $n_L$.

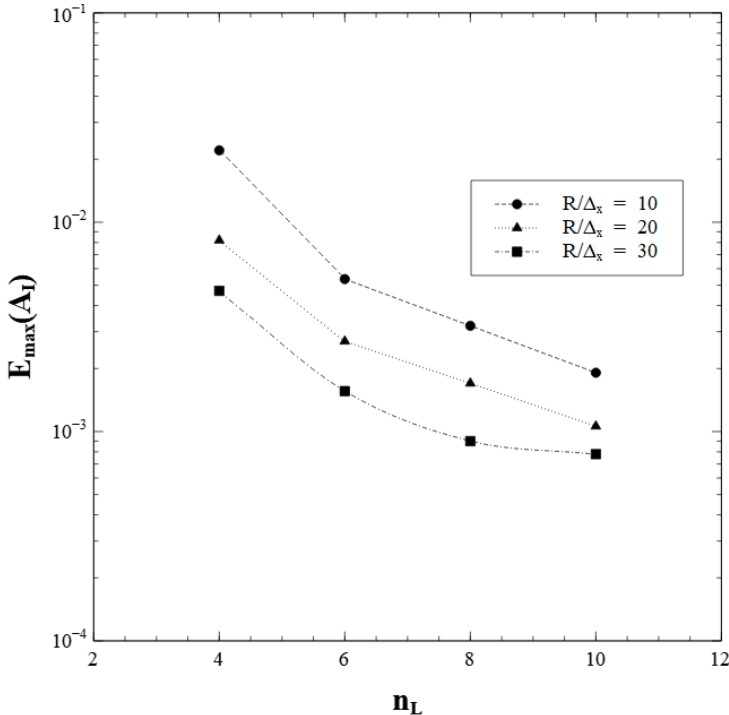

**Figure 8.** Maximum relative interfacial area errors along a circular interface as a function of $n_L$ and the mesh refinement.

## 8. Static Bubble

The analytical solution for the simulation of a stationary bubble in a zero velocity field, and the analytical curvature can be easily obtained from the bubble radius. A circular interface with surface tension should remain at rest, with the pressure jump at the interface exactly balancing the surface tension force (Laplace's law). The velocity field being zero, Equation (4) reduces to:

$$-\nabla \cdot (\tilde{p}\mathbb{I}) = 0 \tag{42}$$

in this test case within each fluid the pressure is constant. The jump relation Equation (9) which is applicable only at the interface reduces to the mathematically exact formulation:

$$[\![p\mathbb{I}]\!] \cdot \mathbf{n}_\mathcal{I} + \gamma\kappa\mathbf{n}_\mathcal{I} = 0 \tag{43}$$

this brings us back to the relation of Laplace. In each phase the pressure is constant and a pressure jump occurs at the interface.

As shown in Figure 9 the pressure jump created by the CSF model at the interface is less direct, which deviates from the real conditions, while the height function methodology gives a better approximation. The points represent the average pressure at the center of the cells and the x-axis represents the distance from the center of the bubble.

In practice, depending on the method used to discretize the pressure gradient and the surface tension force, parasitic currents appear. The exact numerical balance is difficult to obtain [22]. The numerical imbalance created is at the origin of these currents.

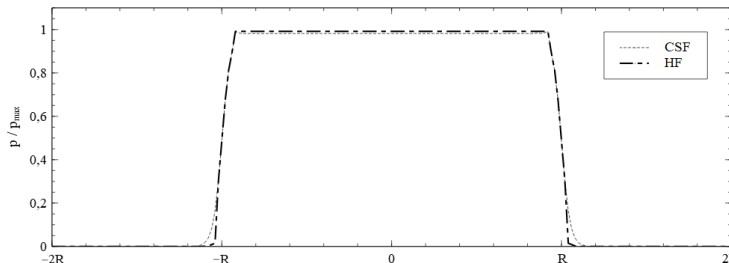

**Figure 9.** Comparison between analytically calculated Young-Laplace pressure, and numerically evaluated pressures around the bubble for the continuous surface force (CSF) model and the height function (HF) model.

Similar to what was done by Popinet [12], it is appropriate to first test the model by imposing the exact curvature in the entire domain for the calculation of $\mathbf{f}_\gamma$. This tests the adequacy of the model by excluding the curvature calculation, and thus verifies that the balance calculation between the pressure term and the surface tension term is indeed achieved. The time required for the momentum to diffuse over a distance $L$ is proportional to $t_\nu$, where

$$t_\nu \propto \frac{L^2}{\nu} \tag{44}$$

and $\nu$ is the kinematic viscosity of the liquid. As noticed by Popinet [12], the time scale needed to reach the numerical equilibrium solution is comparable with the time scale of viscous dissipation $t_\nu$, as expected from physical considerations. In practice, this means that test cases designed to evaluate the accuracy of a given surface tension model (for a stagnant bubble equivalent problem) must ensure that simulations are run for time scales comparable with $t_\nu$. In our case $t_\nu$ is close to 2 ms. The other quantity to consider is the velocity associated with the capillary wave $u_\gamma$.

$$u_\gamma \propto \sqrt{\frac{\gamma}{\rho\,L}} \tag{45}$$

it can be interpreted as the scale of the velocities associated with a capillary wave of wavelength comparable with $L$. As shown in Figure 10, the average velocity obtained converges for a time equivalent to the viscous dissipation time.

The velocity and time have been scaled using $u_\gamma$ and $t_\nu$. Thus, the numerical calculation verifies the theoretical equilibrium and the spurious currents observed in the following can be attributed to errors in the curvature calculation.

In this second test the curvature is calculated by the model. In order to evaluate the impact that spurious currents could have on a simulation with Marangoni effect, the results obtained are scaled using an average speed of Marangoni currents observed in experiments $u_M = 25 \text{ mm} \cdot \text{s}^{-1}$.

$$u^* = \frac{u_{max}}{u_M} \tag{46}$$

as shown in Figure 11, for the CSF method the spurious currents increase when $\Delta x$ decreases. Several simulations were performed for different mesh resolutions for each of the two tested models: CSF and the one based on HF. The simulations were carried out for a time equivalent to the experimentally observed growth time of a bubble.

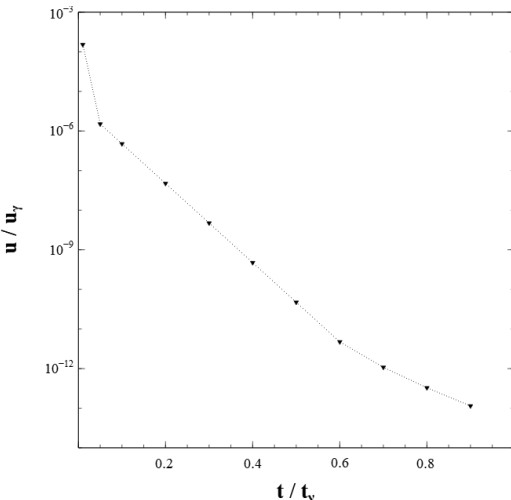

**Figure 10.** Evolution of the maximum intensity of the spurious currents observed around the bubble. With the use of an exact curvature for the simulation, the equilibrium is reached for a time $t_v$.

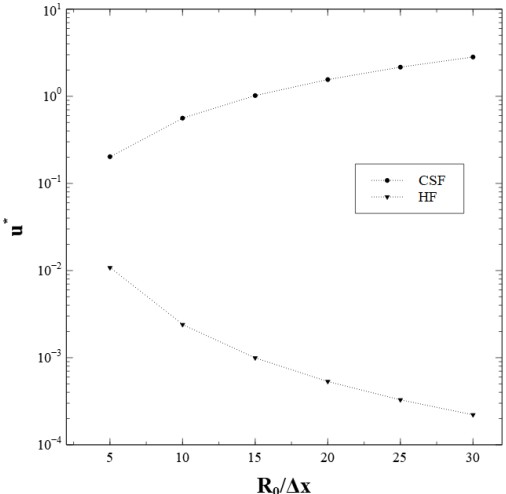

**Figure 11.** Convergence with spatial resolution of maximum spurious currents velocities.

This is consistent with the analysis presented by Harvie et al. [37]. The CSF method is therefore clearly not suitable for the simulation that is the objective of this study. This validates the use of a more efficient interface representation method. The error generated using HF decreases with the reduction of the grid spacing. The results obtained in this section show that the method used is balanced and allows estimation of the curvature sufficiently accurate to obtain a solution close to the exact equilibrium (for the velocity). The numerical equilibrium obtained is very close to the theoretical equilibrium. Even for coarse resolutions the error generated on the final simulation is less than 1%.

## 9. Translating Bubble

While the case of a stagnant bubble allows us to test the equilibrium of the model by referring to an exact solution of the velocity field, it does not allow us to evaluate the combined accuracy of the interface advection and surface tension model. As proposed by Popinet [12], the horizontal translation of a bubble carried by a uniform flow field is a more robust and realistic test. In the case of electrogenerated bubbles, when the bubble grows, the interface moves at a vertical speed of a few millimeters per second. This is a preliminary test before testing mass transfer models across the interface. A uniform horizontal velocity $u_0$ is imposed in the whole domain with periodic boundary conditions on lateral sides and

symmetry boundary conditions on the top and bottom. As already reported by Popinet, the absolute error on the velocity does not depend on $u_0$ and is weakly dependent on the Laplace number $La = \dfrac{\rho L \gamma}{\mu}$. It is thus the transport scheme of the interface that is directly tested and therefore the impact of the spatial resolution. In our study the Laplace number varies between 5000 and 20,000. A new time scale is introduced, to account for the time needed for the bubble to cross a length $L$, and is given by:

$$t_{u_0} \propto \frac{L}{u_0} \tag{47}$$

the velocity has been scaled with $u_M$ and the time with $t_{u_0}$. The Laplace number was fixed at 12,000. The results presented here as an example reflect a general trend in the evolution of the spurious velocity over time as observed by Abadie et al. [20]. Popinet notes that these oscillations are proportional to $u_0/\Delta x$. The advection errors of the models continuously disturb the calculations related to the interface [12].

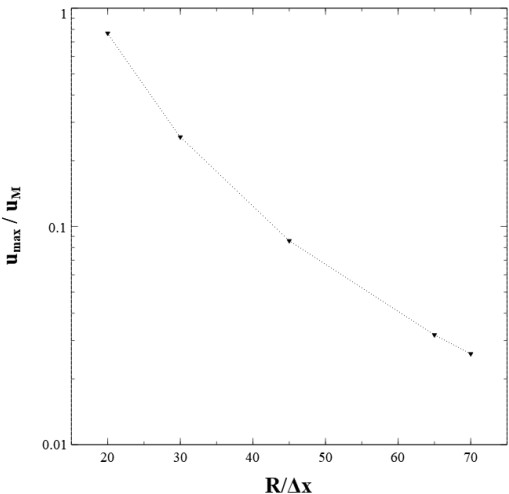

**Figure 12.** Non-dimensionnal spurious currents velocity as a function of spatial resolution.

The model has similar behavior to the previous studies [12,20]. As illustrated in Figure 12, the drastic drop in performance can be noticed. Even if the height function method allows accurate curvature calculations, the flaw of the method comes essentially from the advection method. The previous simulation was repeated for different mesh resolutions. The dimensionless quantity used in the abscissa is $R/\Delta x$. As for the static case the method converges when the mesh is refined. We obtain an error of 2.5% for $R/\Delta x = 70$.

## 10. Surface Gradient Error Calculation

Next, the efficiency of the surface tension gradient calculation should be tested, as shown in Equation (29). The static bubble is subjected to different temperature gradients over a given distance as shown in Figure 13. The value calculated with Equation (29) is compared with the exact value.

The objective here is to expose the interface of the bubble to variations in surface tension similar to what it might encounter as it grows, the bubble is exposed to surface tension variations ranging from $0.1 \, \text{N} \cdot \text{m}^{-2}$ to $50 \, \text{N} \cdot \text{m}^{-2}$. As the interface is circular the exact value of the surface tension gradient can be calculated. For each cell crossed by the interface the length of the interface is known and every two cells the temperature difference can be calculated. For each simulation the surface tension gradient is averaged along the interface in order to compensate for uncertainties concerning the influence of the position

of the interface within the cell. The relative error found is calculated according to the following equation:

$$E(\nabla_s \gamma) = \frac{\sum^N |\nabla_s \gamma_{exact} - \nabla_s \gamma| / |\nabla_s \gamma_{exact}|}{N} \tag{48}$$

where $N$ is the number of cells used for the calculation. The maximum errors found are also recorded, and calculated using the equation:

$$E_{\max}(\nabla_s \gamma) = \frac{\max |\nabla_s \gamma_{exact} - \nabla_s \gamma|}{|\nabla_s \gamma_{exact}|} \tag{49}$$

the errors found are counted and plotted in Figure 14.

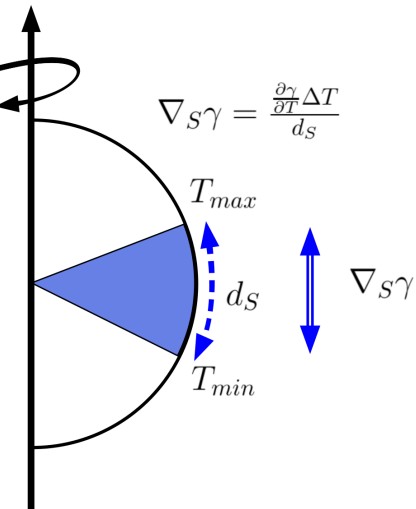

**Figure 13.** Surface gradient calculation.

The finer the mesh, the smaller the error. As shown in the graph in Figure 14, the calculated errors are not very sensitive to the value of the surface tension gradient, but they decrease rapidly as the mesh is refined. The maximum $E_{\max}$ and average $E_{avg}$ error are calculated for two values of the surface tension gradient, $0.1 \text{ N} \cdot \text{m}^{-2}$ and $50 \text{ N} \cdot \text{m}^{-2}$. The maximum errors are mainly due to cells where the interface share is small compared with the total cell volume.

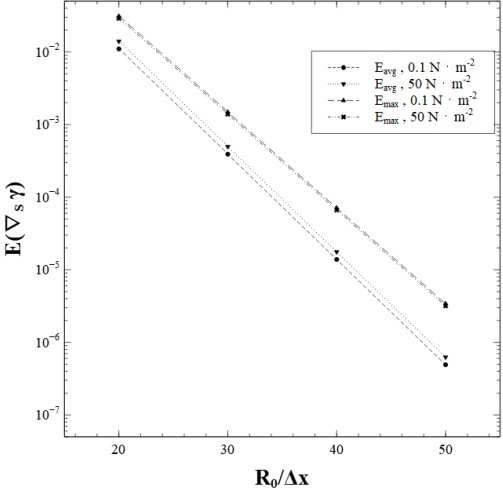

**Figure 14.** Relative error of the surface tension gradient as a function of mesh resolution.

## 11. Conclusions

In summary, a surface tension model based on height functions has been presented. The generated spurious currents were compared with the average Marangoni current expected during the growth of an electrogenerated bubble. This model allows accurate treatment of the surface tension force. For reasonable mesh sizes, the spurious currents typically present when using fixed grids can then be reduced to machine precision. For the case of a stationary bubble, the method shows convergence to the theoretical solution with an increasingly fine spatial resolution. This is not the case for the CSF method. In the case of a translating bubble, a drastic drop in performance can be noticed. However, the spurious currents disappear with a better resolution of the mesh. For a bubble radius greater than 70 times the width of the mesh, an error lower than 2.5% is expected for the simulation of Marangoni currents.

**Author Contributions:** Conceptualization, F.S. and Z.G.; Formal analysis, D.F.F., M.K., R.I., M.S. and P.M.; Methodology, Z.G. and M.S.; Software, Z.G.; Supervision, M.S. and P.M.; Writing—original draft, F.S.; Writing—review and editing, D.F.F., M.K., R.I. All authors have read and agreed to the published version of the manuscript.

**Funding:** This research received no external funding.

**Conflicts of Interest:** The authors declare no conflict of interest.

### Nomenclature

The following nomenclature are used in this manuscript:

| | |
|---|---|
| $\mathbf{n}_\mathcal{I}$ | unit vector normal to the interface |
| $\mathbf{v}$ | velocity vector |
| $\mathbf{v}_\mathcal{I}$ | interfacial velocity vector |
| $\rho$ | density |
| $\mu$ | dynamic viscosity |
| $\dot{m}$ | mass flow rate of mass transfer at the interface |
| $\gamma$ | surface tension |
| $\kappa$ | curvature |
| $\alpha$ | volume fraction |
| $\delta_\mathcal{I}$ | dirac function |
| $\Delta_x$ | width of the mesh cells |
| $\Delta_y$ | height of the mesh cells |
| $A_\mathcal{I}$ | interface area |
| $V_{cell}$ | cell volume of the mesh |
| $\Delta_t$ | time step of the simulation |
| $T$ | temperature |
| $K_H$ | Henry's law constant |
| $c_{h_2}$ | concentration of dissolved hydrogen |
| $c_{h_2}^S$ | saturation concentration of dissolved hydrogen |
| $u_M$ | average velocity of Marangoni currents |
| $\nu$ | kinematic viscosity of the liquid |
| $L$ | characteristic length |
| $t_\nu$ | momentum diffusion time |
| $R$ | radius of the bubble |
| $u_0$ | liquid velocity |
| $u_\gamma$ | capillary wave velocity |

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
