# Peer review of "Suitability of the VOF Approach to Model an Electrogenerated Bubble with Marangoni Micro-Convection Flow"

_fluids, doi:10.3390/fluids7080262_

Round 1

Reviewer 1 Report

This study proposed a novel numerical approach that considered the impact of the Marangoni effect on bubble formation, effects on mass transfer, electrocapillary effect etc. The results show significant improvement compared to previous studies in which such effects have been overlooked. The methodology of the research is solid. I recommend the paper be published with minor revision.

Please refer to my comments below:

·        The authors stated that to study the phenomenon of microconvective vortex flow induced by Marangoni effect numerically, surface tension variation along with the liquid-gas interface, integration of mass transfer, the moving contact line need to be considered. Could you please elaborate on the reason for identifying the above to the interest of this study?

·        The paragraph starts from line 30, describing the impact of force on the evolution of the bubble, which needs more reference.

·        Generally, bubble formation/evolution requires describing the pathway processes such as nucleation, surface growth, agglomerate and coalescence, and breakup. Could you please elaborate more on the numerical representation of the process?

Author Response

Thank you for taking the time to review our paper submission. We have
taken all of your feedback into consideration and have responded to each of
your comments below.

 "The authors stated that to study the phenomenon of microconvective vortex flow induced by Marangoni effect numerically, surface tension variation along with the liquid-gas interface, integration of mass transfer, the moving contact line need to be considered. Could you please elaborate on the reason for identifying the above to the interest of this study?":
The paragraph at line 96 has been added.

"The paragraph starts from line 30, describing the impact of force on the evolution of the bubble, which needs more reference.":
The following reference is added "Taqieddin, A.; Nazari, R.; Rajic, L.; Alshawabkeh, A. Review—Physicochemical Hydrodynamics of Gas Bubbles in Two Phase
559 Electrochemical Systems 2017. 164, E448–E459.".

"Generally, bubble formation/evolution requires describing the pathway processes such as nucleation, surface growth, agglomerate and coalescence, and breakup. Could you please elaborate more on the numerical representation of the process?":
-    Concerning nucleation, bubbles will first form at surface cavities.  From a numerical point of view, when using a VOF method as envisaged in this article, the starting point would be a prescribed shape and size of the bubble. 
-    In order to simulate the growth of the bubble it is necessary to simulate the interfacial mass transfer from the liquid to the gas phase, and see the answer to the first comment.
-     Regarding bubble detachment or coalescence, the VOF method automatically considers the changes in topology. This justifies the choice of this type of method as opposed to a moving mesh method.

Reviewer 2 Report

The referencing style is to be checked and corrected; the number of the reference is to be indicated. For example, in line 50, Yang et al. [6].

The novelty of the study is to be clearly stated.

The advantages and disadvantages of the VOF method are to be detailed.

The title of Fig. 3 is to be reduced; the needed description is to be indicated in the text. (the same for Figs. 9, 11 and 13…)

The results presented in Fig.3 is confusing; is it you results or from the literature? To be explained.

A figure presenting the used mesh is to be added.

A qualitative verification of the numerical model is to be performed.

Some qualitative results are to be presented and discussed.

Some physical interpretations are to be added to the discussion.

How is the mesh treated for the Translating bubble?

The boundary conditions are to be expressed mathematically.

A nomenclature is to be added.

Author Response

Thank you for taking the time to review our paper submission. We have
taken all of your feedback into consideration and have responded to each of
your comments below.

"The referencing style is to be checked and corrected; the number of the reference is to be indicated. For example, in line 50, Yang et al. [6].":
The referencing style has been modified accordingly.

The novelty of the study is to be clearly stated.
A paragraph beginning at line 160 has been added accordingly.

"The advantages and disadvantages of the VOF method are to be detailed.": 
A paragraph beginning at line 127 has been added accordingly.

"The title of Fig. 3 is to be reduced; the needed description is to be indicated in the text. (the same for Figs. 9, 11 and 13…)":
The description is now indicated in the text.

"The results presented in Fig.3 is confusing; is it you results or from the literature? To be explained.":
The results shown in Figure 3 are our results and were obtained using the methodology described in this article.

"A figure presenting the used mesh is to be added."and "How is the mesh treated for the Translating bubble?":
The type of mesh used for the moving bubble is the same as that used for the static bubble, and consists of quadrilaterals of the same size. As the curvature of the interface does not change in either case, the mesh is not locally refined. A description of the mesh is now added to the text line 363 but because it is a simple structured mesh with quadrilateral elements, we did not think that it would be a good use of space to include such an elementary mesh.

"Some physical interpretations are to be added to the discussion.": 
The paragraph at line 96 has been added. However, it seems premature to discuss further the hypotheses presented in this paragraph as we cannot answer them. The purpose of this paragraph is to show the need to develop the numerical methods presented in this paper.

"A nomenclature is to be added.":
A nomenclature has been added.

"A qualitative verification of the numerical model is to be performed."and "Some qualitative results are to be presented and discussed.":
The objective of this paper is to develop numerical methods capable of simulating an electrogenerated bubble. However, for the time being, a contact line model has not yet been integrated, which does not allow the simulation of a bubble attached to the solid surface of an electrode. 
We have demonstrated in the paper that the proposed numerical methods are appropriate to capture the key physical phenomena that occur in electrogenerated bubbles (Marangoni effect, interfacial mass transfer). 
We have also provided a verification by showing that the pressure jump across the interface satisfies Laplace's formula. Two test cases were used to test the validity of the numerical method employed. In addition, errors in the calculation of curvature, interface density, and surface tension gradient were tested independently. 
The qualitative results we have presented concern the effect of the mesh density on the quality of the numerical method. The suitability of the methodology used in the VOF framework to simulate an electrogenerated bubble, i.e. for low capillary numbers and with a high density ratio was tested, which is the main objective of this study. 

Round 2

Reviewer 2 Report

After revision, the paper can be accepted for publication